# Compressing deep graph convolution network with multi-staged knowledge distillation

**Junghun Kim[1], Jinhong Jung[2], U. Kang[1]***

**1** Seoul National University, Seoul, Republic of Korea, **2** Jeonbuk National University, Jeonju-si, Jeollabuk-do, Republic of Korea

* ukang@snu.ac.kr

**Data Availability Statement:** The source code and the datasets are available at https://github.com/snudatalab/MustaD.

**Funding:** This work was supported in part by IITP grant funded by the Korea government [No. 2021-

## Abstract

Given a trained deep graph convolution network (GCN), how can we effectively compress it into a compact network without significant loss of accuracy? Compressing a trained deep GCN into a compact GCN is of great importance for implementing the model to environments such as mobile or embedded systems, which have limited computing resources. However, previous works for compressing deep GCNs do not consider the multi-hop aggregation of the deep GCNs, though it is the main purpose for their multiple GCN layers. In this work, we propose MustaD (Multi-staged knowledge Distillation), a novel approach for compressing deep GCNs to single-layered GCNs through multi-staged knowledge distillation (KD). MustaD distills the knowledge of 1) the aggregation from multiple GCN layers as well as 2) task prediction while preserving the multi-hop feature aggregation of deep GCNs by a single effective layer. Extensive experiments on four real-world datasets show that MustaD provides the state-of-the-art performance compared to other KD based methods. Specifically, MustaD presents up to 4.21%p improvement of accuracy compared to the second-best KD models.

## Introduction

*Given a trained deep graph convolution network, how can we compress it into a compact network without a significant drop in accuracy?* Graph Convolution Network (GCN) [1] learns latent node representations in graph data, and plays a crucial role as a feature extractor when a model is jointly trained to learn node features and perform a specific task. GCN has attracted considerable attention from research community because it enables researchers to easily and effectively analyze graphs. Various GCN models [2–4] have been proposed to boost the performance of tasks on real-world graphs such as node and graph classification [1], link prediction [5], relation reasoning [6], etc.

Recently, the research on deep-layered GCNs is highly in progress to extract sophisticated node features in large and complicated graphs [7–13]. Those deep GCN models have many layers to understand patterns of large graphs better and improve their performance. However,

0-01343, Artificial Intelligence Graduate School Program (Seoul National University)], in part by Institute of Information & Communications Technology Planning & Evaluation (IITP) grant funded by the Korea government(MSIT) (No.2020-0-00894, Flexible and Efficient Model Compression Method for Various Applications and Environments), and in part by the ICT R&D program of MSIT/IITP (No.2017-0-01772, Development of QA systems for Video Story Understanding to pass the Video Turing Test). The Institute of Engineering Research and ICT at Seoul National University provided research facilities for this work. The funders had no role in study design, data collection and analysis, decision to publish, or preparation of the manuscript.

**Competing interests:** The authors have declared that no competing interests exist.

as the number of layers increases, the number of parameters to be trained also increases, and this leads to a non-negligible increase of model size. Therefore, it is difficult to use those large models in environments having limited computing resources such as mobile or embedded systems.

Model compression aims to learn compressed and lightweight deep networks for low-powered and resource-limited devices without a significant loss of predictive accuracy. For the purpose, many researchers have proposed various strategies such as parameter pruning [14], low-rank factorization [15], weight quantization [16], and knowledge distillation [17]. Among them, Knowledge Distillation (KD) has been popular due to its simplicity based on a student-teacher model; KD distills the knowledge from a large teacher model into a smaller student model so that the student performs as well as the teacher [18–20]. In this context, Yang et al. [21] have recently proposed a KD method called LSP (Local Structure Preserving) for compressing GCN models. However, LSP deals with rather shallow models, and only distills limited knowledge on feature aggregation of a teacher while disregarding various aspects to be considered when a network becomes deep. Specifically, LSP does not consider the teacher's knowledge on multi-hop feature aggregation although the process is essentially involved in a deep-layered GCN; thus, its performance on preserving accuracy is limited, especially for compressing a deep GCN.

In this paper, we propose MustaD (Multi-staged knowledge Distillation), a novel approach for compressing deep GCNs to single-layered GCNs through multi-staged knowledge distillation (KD) while preserving the multi-hop feature aggregation of deep GCNs. Based on the concept of knowledge distillation, MustaD aims to train a single-layered student GCN with the same or even a lower feature dimension than that of a trained teacher GCN. The framework of MustaD is illustrated in Fig 1. Our main idea is to distill the knowledge of multi-hop feature aggregation from multiple GCN layers as well as that of task prediction. Specifically, the single-layered student learns the knowledge of multi-hop feature aggregation of the teacher by 1) matching hidden feature embeddings from the teacher, and by 2) imitating the multiple GCN layers of the teacher with a single effective layer. The knowledge of task prediction is distilled to the student by transferring the probabilistic prediction vector of the teacher. These multi-staged knowledge distillations guide the student to obtain similar aggregated features and predictions to the deep-layered teacher with significantly less parameters.

Fig 2 depicts the overall performance of our MustaD compared to other KD-based methods. Our proposed method Student_MustaD shows the best performance among KD methods, especially for deep teachers.

Our contributions are summarized as follows:

- **Method**. We propose MustaD, a novel approach for compressing deep-layered GCNs through distilling the knowledge of both the feature aggregation and the feature representation. We propose a simple but powerful method to preserve the multi-hop feature aggregation of the teacher with significantly less parameters.

- **Theory**. We provide theoretical analysis of the proposed MustaD, and show that the expressiveness of the student from MustaD is similar to that of a deep-layered GCN on a spectral domain.

- **Experiment**. We validate MustaD on two trained deep GCN models in four datasets compared to other distillation-based GCN compression methods. In particular, we improve the accuracy by 3.95%p, 3.77%p, 4.21%p compared to the second-best KD models on Cora, Citeseer, and Pubmed, respectively. In ogbn-proteins, MustaD presents an 1.55%p improvement in terms of AUC-ROC from the second-best KD model.

### Teacher Model

### Student Model

**Fig 1. Framework of MustaD.** MustaD preserves the multi-hop feature aggregation of a teacher with a single effective layer in a student. Furthermore, MustaD distills knowledge of 1) aggregation from multi-staged GCN layers as well as 2) task prediction. $\mathbf{h}_{i;t}$ represents the teacher's last hidden embedding of node $i$, and $\tilde{\mathbf{h}}_{i;s}$ corresponds to the student's last hidden embedding of node $i$ where the hidden dimension is matched to the teacher. $\mathbf{p}_{i;t}$ and $\tilde{\mathbf{p}}_{i;s}$ denote the prediction probability vectors of node $i$ of the teacher, and the student, respectively.

The code and the datasets are available at https://github.com/snudatalab/MustaD.

## Related work

Many complex and deep networks are proposed to solve real-world tasks such as text classification [22], malware detection [23], in-vehicle intrusion attack detection [24], and web

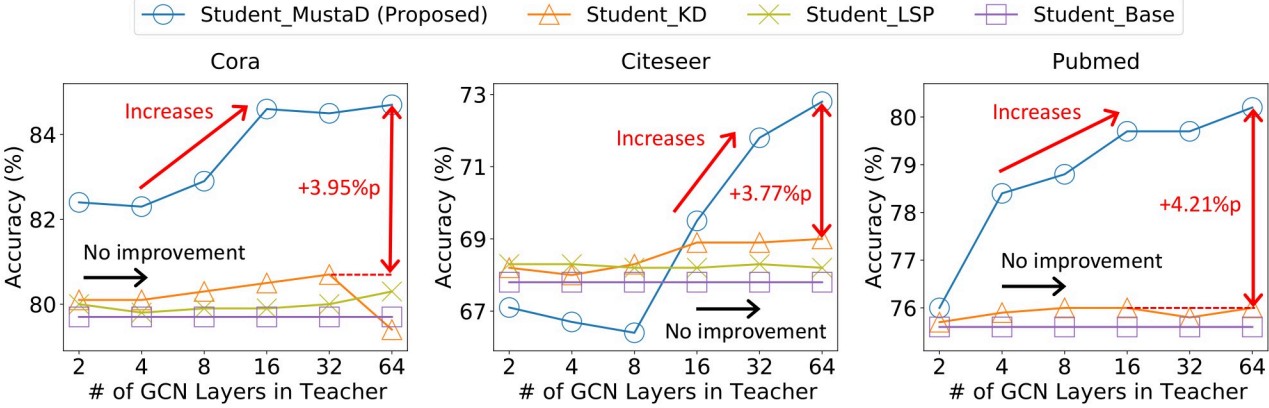

**Fig 2. Accuracy of student models for different number of GCN layers in a teacher model.** Student_KD and Student_LSP represent the students trained by distilling the knowledge of classes, and knowledge of the embedded topological structure of a teacher, respectively. Student_Base corresponds to a model trained with the ground truth labels without the teacher. Note that our proposed MustaD (denoted as Student_MustaD) provides the highest accuracy in most cases. We also observe that MustaD provides much better performance for deep GCN with many layers, unlike competitors whose performances do not improve with more layers.

document classification [25]. In particular, several deep graph convolutional networks (GCNs) are proposed to handle real-world graphs [7–13, 26, 27]. However, it is difficult to use these models in environments with limited computing resources. Therefore, many Knowledge Distillation (KD) methods have been studied to compress a large teacher model to a smaller student model by extracting compact and useful information [17, 18, 28–30]. Although those methods improve the efficiency of compression, they are designed for the data in a grid domain only; it is hard for them to be directly applied to the data in the non-grid domain such as graphs.

In this section, we discuss related works on deep GCN and KD methods. Table 1 summarizes the symbols used in this paper.

## Deep graph convolution network

Since the first GCN has been proposed in [1], many convolution based graph neural networks are proposed [2–4]. In GCNs, a convolution layer aggregates feature information from one-hop neighbors, and multiple convolution layers aggregate feature information from multi-hop neighbors. Recently, many deep GCNs are studied to consider the multi-hop feature information [9–12, 27].

ResGCN [7] borrows residual/dense connections and dilated convolutions from CNNs, and adapts them to GCN architectures. GEN [8] is a complementary version of [7]. The model uses a modified graph skip connection which is a pre-activation version of residual connections in ResGCN.

**Table 1. Table of symbols.**

| Symbol | Definition |
|---|---|
| $\mathcal{G} = (\mathcal{V}, \mathcal{E})$ | input graph. $\mathcal{V}$: node set, $\mathcal{E}$: edge set. |
| $N$ | number of nodes. |
| $d$ | input feature dimension. |
| $\mathbf{X} \in \mathbb{R}^{N \times d}$ | input feature matrix. |
| $\mathbf{x}_i \in \mathbb{R}^d$ | input feature vector for node $i$. |
| $\mathbf{H}^{(l)} \in \mathbb{R}^{N \times d}$ | hidden feature embedding matrix of $l$-th GCN layer; vector $\mathbf{h}_i^{(l)} \in \mathbb{R}^d$ of $i$-th row of $\mathbf{H}^{(l)}$ represents that for node $i$. |
| $\mathbf{h}_{i;t} \in \mathbb{R}^d$ | teacher's hidden embedding of node $i$. |
| $\mathbf{h}_{i;s} \in \mathbb{R}^d$ | student's hidden embedding of node $i$. |
| $\mathcal{N}_i$ | set of one-hop neighbors of node $i$ in $\mathcal{G}$. |
| Emb(·) | learnable function that maps a given feature onto a new embedding space. |
| Aggregation (·) | aggregation function that aggregates hidden features from one-hop neighbors. |
| $K$ | number of layers. |
| GCN$_s$(·) | single effective GCN layer in MUSTAD; shared in the student model. |
| $\mathcal{K}(\cdot)$ | kernel function. |
| $\mathcal{D}_{KL}(\cdot)$ | Kullback-Leibler divergence. |
| $\mathbf{p}_{i;s}$ | prediction probability vector of the student. |
| $\mathbf{p}_{i;t}$ | prediction probability vector of the teacher. |
| $\lambda_{emb}$ | hyperparameter for the embedding loss. |
| $\lambda_{pred}$ | hyperparameter for the prediction loss. |
| $\alpha$ | hyperparameter for the initial residual. |
| $\beta$ | hyperparameter for the identity mapping. |

GCNII [13] extends the vanilla GCN model to overcome the over-smoothing problem proposed in [31]; [31] observes that given a renormalized graph convolution matrix $\tilde{\mathbf{P}}$ and an input feature matrix $\mathbf{X}$, a $K$-layer vanilla GCN simulates a fixed $K$-order polynomial filter $\tilde{\mathbf{P}}^K\mathbf{X}$, and the over-smoothing problem is caused when $\tilde{\mathbf{P}}^K\mathbf{X}$ converges to a distribution that does not carry the information of $\mathbf{X}$. To overcome the over-smoothing problem, GCNII introduces initial residual and identity mapping techniques to the vanilla GCN. The initial residual constructs a skip connection from the input layer, thus ensuring that the final representation of each node retains at least a fraction of $\mathbf{X}$. The identity mapping merely transfers the aggregated features to the next GCN layer without any parameterized embedding process. Each GCNII layer is characterized as:

$$\mathbf{H}^{(l+1)} = \sigma\big(\big((1 - \alpha_{l+1})\tilde{\mathbf{D}}^{-\frac{1}{2}}\tilde{\mathbf{A}}\tilde{\mathbf{D}}^{-\frac{1}{2}}\mathbf{H}^{(l)} + \alpha_{l+1}\mathbf{X}\big)\big((1 - \beta_{l+1})\mathbf{I}_N + \beta_{l+1}\mathbf{W}^{(l+1)}\big)\big) \tag{1}$$

where $\mathbf{H}^{(l)}$ corresponds to the $l^{\text{th}}$ hidden feature representation, $\tilde{\mathbf{A}}$ represents the normalized adjacency matrix, $\tilde{\mathbf{D}}$ represents the degree matrix of $\tilde{\mathbf{A}}$, $\mathbf{I}_N$ denotes the identity matrix, and $\sigma$ denotes the activation function. $\alpha_{l+1}$ and $\beta_{l+1}$ are two hyperparameters where $\alpha_{l+1}$ controls the power of connection for the initial feature $\mathbf{X}$ to $(l + 1)^{\text{th}}$ GCN layer, and $\beta_{l+1}$ controls the degree of merely transferring the aggregated features to the next GCN layer without any parameterized embedding.

Although many deep GCN models accelerate their performance by considering the multi-hop features in graphs, it is difficult to use them in environments with limited computing resources such as mobile or embedded systems due to their large model sizes. In this paper, we concentrate on compressing a deep GCN into a shallow GCN while preserving the multi-hop feature aggregation property of deep GCNs.

## Knowledge distillation

Knowledge Distillation (KD) [17] transfers knowledge from a large teacher model into a smaller student model so that the student performs as well as the teacher. In the method, task predictions of a teacher is smoothed by the softmax function. Distillation of knowledge is done by making task predictions of the student be similar to that of the teacher. Several KD methods distill not only the output of teachers but also the information of intermediate hidden layers [18, 19]. [20] introduces intermediate-level hints from hidden layers of a teacher to guide a student to learn intermediate representations of the teacher. However, those methods aims to compress a wide and *shallow* teacher model into a thin and shallow student model; i.e., they do not focus on compressing a *deep* teacher GCN model into a shallow GCN model. Thus, they have limitations in compressing multiple GCN layers into few GCN layers.

Recently, to our best knowledge, the first KD method on GCNs based on Local Structure Preserving (LSP) module is proposed in [21]. In the module, topological semantics from both the teacher and the student are extracted as distributions, and the topology-aware knowledge transfer is done by minimizing the distance between these distributions. However, LSP only transfers the intermediate knowledge not considering task predictions which is specially designed for the objective task. Furthermore, LSP does not consider the teacher's knowledge on multi-hop feature aggregation in a student although the process is essentially involved in a deep GCN. Therefore, its performance about preserving the accuracy is limited, especially for compressing a deep GCN.

## Proposed method

In this section, we propose MUSTAD (Multi-Staged Knowledge distillation), a novel approach for effectively compressing a deep GCN by distilling multi-staged knowledge from a teacher.

We summarize the challenges and our ideas in developing our distillation method while preserving the multi-hop feature aggregation of the deep-layered teacher.

1. When compressing a deep teacher GCN model to a small student GCN model by distilling knowledge from the teacher model, it is essential to conserve the multi-hop feature aggregation of the deep model as the aggregation is the key purpose of stacking multiple GCN layers. We propose to use a **single effective layer** that imitates the *K* GCN layers in the teacher model by a single GCN layer in the student while preserving the multi-hop feature aggregation process and reducing the model size significantly.

2. It is also important to decide what knowledge to be distilled to preserve the performance of the teacher model in the student model. We propose **multi-staged knowledge distillation** that distills not only the knowledge of the teacher model's task predictions but also its final hidden embeddings to the student model. By distilling the knowledge of the final hidden embeddings, the student model generates its final representation similar to that of the teacher model; thus, the multi-staged knowledge distillation helps the single effective layer imitate the multiple GCN layers.

Firstly, we describe how to preserve multi-hop feature aggregation of the teacher model in a single effective student network based on the observation of the fundamental mechanism of deep GCNs. Then we describe the knowledge distillation of embeddings as well as task predictions, followed by the explanation of the final loss function for jointly training all of them for the node classification task. At last, we give a spectral analysis of MUSTAD when distilling the knowledge of GCNII teacher model to strengthen the theoretical background of our method.

### Preserving multi-hop feature aggregation

We describe how MUSTAD preserves the feature aggregation procedure of deep GCN layers of the teacher in a single GCN layer of the student. The main purpose of deep GCN is to consider multi-hop neighbors using multiple GCN layers. Let $\mathcal{G} = (\mathcal{V}, \mathcal{E})$ denote an input graph where $\mathcal{V}$ and $\mathcal{E}$ denote the sets of nodes and edges, respectively. Given the graph $\mathcal{G}$, a GCN layer is expressed by

$$\mathbf{h}_i^{(k+1)} = \text{GCN}^{(k+1)}(\mathbf{h}_i^{(k)}) := \underset{j \in \mathcal{N}_i \cup i}{\text{Aggregation}}(\text{Emb}_{k+1}(\mathbf{h}_j^{(k)})) \qquad (2)$$

where $\mathbf{h}_i^{(k)}$ denotes the hidden feature embedding for node $i$ in the $k$-th GCN layer, $\mathcal{N}_i$ denotes the set of one-hop neighbors of node $i$ in $\mathcal{G}$, and $\text{Emb}_k(\cdot)$ is a learnable function that maps a given feature onto a new embedding space, which is used in the $k$-th GCN layer. According to Eq (2), a GCN layer aggregates hidden features from one-hop neighbors to obtain new hidden features by Aggregation(·). Thus, when a model uses $K$ GCN layers, it aggregates hidden features from up to $K$-hop neighbors.

Given a teacher model having $K$ GCN layers, our MUSTAD preserves the process by imitating the teacher's multi-hop feature aggregation in a single effective layer which is represented by the following equation:

$$\mathbf{h}_i^{(k+1)} = \text{GCN}_s(\mathbf{h}_i^{(k)}) \text{ for } k = 1, 2, \cdots, K \qquad (3)$$

where $\text{GCN}_s(\cdot)$ indicates a shared GCN layer in the student model, and $\mathbf{h}_i^{(k)}$ denotes the hidden

embedding of node $i$ at $k$-th iteration in the student. In other words, MustaD repeats $\text{GCN}_s(\cdot)$ $K$ times in the student model to imitate the teacher's multi-hop aggregation as shown in Fig 1. Thus, our model reduces the number of model parameters by compressing multiple GCN layers into a single layer while effectively considering multi-hop feature aggregation.

## Distilling knowledge from trained deep GCNs

MustaD distills the teacher's multi-staged knowledge of embeddings and task predictions to the student as depicted in Fig 1.

**Distilling knowledge of embeddings.** MustaD distills the last hidden embeddings after $K$-hop aggregations of the teacher into the student. This distillation guides the student to follow the teacher's behavior more carefully. The main idea for the distillation is to make embeddings of both the teacher and the student similar by minimizing the following loss function:

$$\mathcal{L}_{emb} = \operatorname*{mean}_{i \in \mathcal{V}}(\mathcal{K}(\tilde{\mathbf{h}}_{i;s}, \mathbf{h}_{i;t})) \tag{4}$$

where $\mathbf{h}_{i;t}$ is the teacher's last hidden embedding of node $i$, $\tilde{\mathbf{h}}_{i;s} = \mathbf{W}_s \mathbf{h}_{i;s}$ where $\mathbf{h}_{i;s}$ is the student's last embedding of node $i$, and $\mathbf{W}_s$ is a learnable weight matrix used to match the dimension between the teacher and the student. The matching layer is omitted if they have the same hidden dimension. $\mathcal{K}(\cdot)$ is a kernel function to measure the distance between the two given embedding vectors, and any distance metric can be used. In this work, we investigate the effect of kernel functions among the following metrics:

$$\mathcal{K}(\tilde{\mathbf{h}}_{i;s}, \mathbf{h}_{i;t}) = \begin{cases} \|\tilde{\mathbf{h}}_{i;s} - \mathbf{h}_{i;t}\|_p & (\text{Distance}-\text{based kernel}) \\[2mm] \tilde{\mathbf{h}}_{i;s}^\top \mathbf{h}_{i;t} & (\text{Linear kernel}) \\[2mm] (\tilde{\mathbf{h}}_{i;s}^\top \mathbf{h}_{i;t} + c)^d & (\text{Polynomial kernel}) \\[2mm] \exp\left(-\frac{\|\tilde{\mathbf{h}}_{i;s} - \mathbf{h}_{i;t}\|_2^2}{2\sigma^2}\right) & (\text{RBF kernel}) \\[4mm] \sum_j \tilde{\mathbf{h}}'_{i,j;s} \log\left(\frac{\tilde{\mathbf{h}}'_{i,j;s}}{\mathbf{h}'_{i,j;t}}\right) & (\text{KL divergence}-\text{based kernel}) \end{cases} \tag{5}$$

where $\mathbf{h}'_{i,j;t}$ and $\mathbf{h}'_{i,j;s}$ denote the $j$-th element of $\mathbf{h}'_{i;t} = \text{Softmax}(\mathbf{h}_{i;t})$ and $\mathbf{h}'_{i;s} = \text{Softmax}(\mathbf{h}_{i;s})$, respectively.

**Distilling knowledge of predictions.** Distilling the knowledge of task predictions follows the process proposed in [17] that minimizes the following loss function:

$$\mathcal{L}_{pred} = \operatorname*{mean}_{i \in \mathcal{V}}(\mathcal{D}_{KL}(\mathbf{p}_{i;s}\|\mathbf{p}_{i;t})) \tag{6}$$

where $\mathcal{D}_{KL}(\cdot)$ is the Kullback–Leibler divergence, $\mathbf{p}_{i;s}$ denotes the prediction probability vector of the student after passing through a softmax function, and $\mathbf{p}_{i;t}$ denotes that of the teacher after passing through a softmax function conditioned with temperature $T$ [17]. The distillation of task prediction guides the student to obtain similar predictive outputs as the teacher.

## Final loss function for node classification

The student model aims to solve the node classification task like the teacher model does. Thus, the student model directly learns the task as well as the aforementioned distillations by

minimizing the following cross entropy loss:

$$\mathcal{L}_{ce} = -\sum_{i \in \mathcal{V}^*} \sum_{j \in \mathcal{C}} y_{ij} \log p_{ij;s} \tag{7}$$

where $\mathcal{V}^*$ is the set of nodes with labels, and $\mathcal{C}$ is the set of labels. $y_{ij}$ is an indicator that is 1 if a node $i$ belongs to label $j$, and 0 otherwise. $p_{ij;s}$ is the probability that the node $i$ belongs to a label $j$, which is predicted by the student. Note that Eq (7) assumes that each node belongs to only one class. If a node has multiple labels (i.e., multi-labeled node classification), we use binary cross entropy loss instead.

To jointly train for all of the aforementioned aspects, MᴜꜱᴛᴀD minimizes the following final loss:

$$\mathcal{L} = \mathcal{L}_{ce} + \lambda_{emb}\mathcal{L}_{emb} + \lambda_{pred}\mathcal{L}_{pred} \tag{8}$$

where $\lambda_{pred}$ and $\lambda_{emb}$ are hyperparameters to balance the proposed loss terms.

## Spectral analysis of MustaD

Spectral graph methods have become fundamental tools in the analysis of large networks [32–34]. GCN [1] has attracted a lot of attention due to its successful implementation of graph convolution defined on a spectral domain as a simple matrix multiplication, thus achieving superior performance compared to other models. In this section, we first give a brief interpretation of $K$-layer GCN on the spectral domain. Then we give a spectral analysis of MᴜꜱᴛᴀD when distilling the knowledge of GCNII teacher model, comparing the expressiveness of our MᴜꜱᴛᴀD to that of $K$-layer GCN on the spectral domain.

Consider an adjacency matrix $\tilde{\mathbf{A}} \in \mathbb{R}^{N \times N}$ of a graph with self-loop, and a graph signal $\mathbf{x} \in \mathbb{R}^N$ which is a set of values residing on a set of nodes, where $N$ is the number of nodes. A polynomial filter of order $K$ on the graph signal $\mathbf{x}$ is defined as

$$K - \text{order polynomial filter on } \mathbf{x} = \left(\sum_{k=0}^{K} \theta_k \tilde{\mathbf{L}}^k\right)\mathbf{x}. \tag{9}$$

where $\tilde{\mathbf{L}} = \mathbf{I}_N - \tilde{\mathbf{D}}^{-1/2}\tilde{\mathbf{A}}\tilde{\mathbf{D}}^{-1/2}$ is the normalized Laplacian matrix of $\tilde{\mathbf{A}}$, and $\theta_l \in \mathbb{R}$ is the polynomial coefficient. $\tilde{\mathbf{D}}$ and $\mathbf{I}_N \in \mathbb{R}^{N \times N}$ represent the degree matrix of $\tilde{\mathbf{A}}$ and the identity matrix, respectively. [31] proves that a $K$-layer GCN simulates a polynomial filter of order $K$ with dependent coefficients $\theta_l$'s, which is the interpretation of $K$-layer GCN on the spectral domain. We show that the student distilled by our proposed MᴜꜱᴛᴀD also simulates the $K$-order polynomial filter with inter-dependent coefficients using only a linear transformation layer and a single effective layer, therefore has a similar expressiveness to the $K$-layer GCN.

Each layer of a teacher that uses GCNII architecture is represented as follows:

$$\mathbf{H}^{(l+1)} = \sigma\left(\left((1 - \alpha_{l+1})\tilde{\mathbf{D}}^{-\frac{1}{2}}\tilde{\mathbf{A}}\tilde{\mathbf{D}}^{-\frac{1}{2}}\mathbf{H}^{(l)} + \alpha_{l+1}\mathbf{X}\right)\left((1 - \beta_{l+1})\mathbf{I}_d + \beta_{l+1}\mathbf{W}^{(l+1)}\right)\right) \tag{10}$$

where $\mathbf{X} \in \mathbb{R}^{N \times d}$ and $\sigma$ denote the input feature matrix and the activation function (ReLU), respectively. $\alpha_{l+1} \in \mathbb{R}$ and $\beta_{l+1} \in \mathbb{R}$ are two hyperparameters. $\mathbf{W}^{(l+1)} \in \mathbb{R}^{d \times d}$ represents a learnable weight matrix in the $(l+1)^{\text{th}}$ GCN layer. $\mathbf{H}^{(l)} \in \mathbb{R}^{N \times d}$ corresponds to the $l^{\text{th}}$ hidden feature representation; i.e., each node has a hidden feature vector of length $d$. The initial hidden representation $\mathbf{H}^{(0)}$ is obtained by a linear transformation of $\mathbf{X}$, expressed by $\mathbf{H}^{(0)} = \mathbf{X}\,\mathbf{W}^{(0)}$. Note that the dimensions of hidden representations for every GCN layers are the same as that of the initial feature vector since there is a residual connection to the input feature matrix $\mathbf{X}$.

As we are dealing with a graph signal $\mathbf{x} \in \mathbb{R}^N$ instead of the input feature matrix $\mathbf{X}$, Eq (10) changes to

$$
\begin{aligned}
\mathbf{h}^{(l+1)} &= \sigma\big(\big((1 - \alpha_{l+1})\tilde{\mathbf{D}}^{-\frac{1}{2}}\tilde{\mathbf{A}}\tilde{\mathbf{D}}^{-\frac{1}{2}}\mathbf{h}^{(l)} + \alpha_{l+1}\mathbf{x}\big)\big((1 - \beta_{l+1}) + \beta_{l+1}w_{l+1}\big)\big) \\
&= \sigma\big(\big((1 - \alpha_{l+1})\tilde{\mathbf{D}}^{-\frac{1}{2}}\tilde{\mathbf{A}}\tilde{\mathbf{D}}^{-\frac{1}{2}}\mathbf{h}^{(l)} + \alpha_{l+1}\mathbf{x}\big)\gamma'_{l+1}\big).
\end{aligned}
\tag{11}
$$

where $w_{l+1} \in \mathbb{R}$ is a learnable parameter, $\gamma'_{l+1} = (1 - \beta_{l+1}) + \beta_{l+1}w_{l+1}$, and $\mathbf{h}^{(l)} \in \mathbb{R}^N$ represents the $l^{\text{th}}$ hidden feature representation; i.e., each node has a hidden representation of length 1. The initial hidden representation $\mathbf{h}^{(0)}$ is obtained by a linear transform of $\mathbf{x}$ which is expressed by $\mathbf{h}^{(0)} = \mathbf{x}\, w_0$.

**Theorem 1** *Consider a K-layer GCNII teacher model. A student of the teacher distilled by MUSTAD expresses a K-order polynomial filter $(\sum_{k=0}^{K} \theta_k \tilde{\mathbf{L}}^k)$ with inter-dependent coefficients $\theta_k$'s for $k \in \{0, \cdots, K\}$ in the following simple form*

$$
\theta_k = \begin{cases}
\gamma(-\gamma)^k - \sum_{s=k+1}^{K} \theta_s \binom{s}{k} & \text{where } k \in \{0, 1, \cdots, K-1\} \\
w_0(-\gamma)^k & \text{where } k = K.
\end{cases}
\tag{12}
$$

**Proof**. We consider a weaker version of the teacher model used in [13], by assuming $\mathbf{x}$ of the signal vector to be non-negative and $\alpha_{l+1} = 1/2$. Furthermore, we remove the ReLU operation since the input feature $\mathbf{x}$ is non-negative as denoted in [13]. Thus, Eq (11) is simplified to the following:

$$
\begin{aligned}
\mathbf{h}^{(l+1)} &= \sigma\big(\big(\tilde{\mathbf{D}}^{-\frac{1}{2}}\tilde{\mathbf{A}}\tilde{\mathbf{D}}^{-\frac{1}{2}}\mathbf{h}^{(l)} + \mathbf{x}\big)\gamma_{l+1}\big) \\
&= \gamma_{l+1}\big(\tilde{\mathbf{D}}^{-\frac{1}{2}}\tilde{\mathbf{A}}\tilde{\mathbf{D}}^{-\frac{1}{2}}\mathbf{h}^{(l)} + \mathbf{x}\big) \\
&= \gamma_{l+1}\big((\mathbf{I}_N - \tilde{\mathbf{L}})\mathbf{h}^{(l)} + \mathbf{x}\big)
\end{aligned}
\tag{13}
$$

where $\gamma_{l+1} = \gamma'_{l+1}/2$, and $\tilde{\mathbf{L}}$ is a normalized Laplacian matrix of the adjacency matrix $\tilde{\mathbf{A}}$. Since MUSTAD uses the repeated single effective layer instead of the $K$ discrete GCN layers, we set $\gamma_{l+1}$'s to a single parameter $\gamma$ for $l \in \{0, \cdots, K-1\}$. Consequently, recursive computations of Eq (13) yield $\mathbf{h}^{(K)}$ of the final representation from the single effective layer as follows:

$$
\mathbf{h}^{(K)} = \big(\sum_{l=0}^{K}\big(\prod_{k=K-l}^{K}\gamma_k\big)(\mathbf{I}_N - \tilde{\mathbf{L}})^l\big)\mathbf{x}
\tag{14}
$$

where $\gamma_k = \gamma$ for $k \in \{1, \cdots, K\}$, and $\gamma_0 = w_0$.

On the other hand, a $K$-order polynomial filter of an adjacency matrix $\tilde{\mathbf{A}}$ on a graph signal $\mathbf{x}$ is expressed by the equation below:

$$
\begin{aligned}
\big(\sum_{k=0}^{K}\theta_k\tilde{\mathbf{L}}\big)\mathbf{x} &= \big(\sum_{k=0}^{K}\theta_k(\mathbf{I}_N - (\mathbf{I}_N - \tilde{\mathbf{L}}))^k\big)\mathbf{x} \\
&= \big(\sum_{k=0}^{K}\theta_k\big(\sum_{l=0}^{k}(-1)^l\binom{k}{l}(\mathbf{I}_N - \tilde{\mathbf{L}})^l\big)\big)\mathbf{x} \\
&= \big(\sum_{l=0}^{K}\big(\sum_{k=l}^{K}\theta_k(-1)^l\binom{k}{l}\big)(\mathbf{I}_N - \tilde{\mathbf{L}})^l\big)\mathbf{x}.
\end{aligned}
\tag{15}
$$

To show that the student of a $K$-layer GCNII teacher distilled by MUSTAD expresses a $K$-order polynomial filter with inter-dependent coefficients, we prove that all $\theta_k$'s for $k \in \{0, 1, \cdots, K\}$ in Eq (15) are expressed by $w_0$ and $\gamma$. Specifically, we show that all $\theta_k$'s in the following equation

$$\prod_{k=K-l}^{K} \gamma_k = \sum_{k=l}^{K} \theta_k (-1)^l \binom{k}{l} \tag{16}$$

are expressed by $w_0$ and $\gamma$ for all $k \in \{0, 1, \cdots, K\}$ where $\gamma_k = \gamma$ for $k \in \{1, \cdots, K\}$, $\gamma_0 = w_0$, and $l \in \{0, 1, \cdots, K\}$. When $l = K$, $\theta_K$ is expressed by $w_0$ and $\gamma$ as follows:

$$\theta_K = w_0 (-\gamma)^K. \tag{17}$$

Recursive computations express all $\theta_k$'s by $w_0$ and $\gamma$ as follows:

$$\theta_{K-1} = \gamma(-\gamma)^{K-1} - \theta_K \binom{K}{K-1}$$

$$\theta_{K-2} = \gamma(-\gamma)^{K-2} - \theta_{K-1} \binom{K-1}{K-2} - \theta_K \binom{K}{K-2} \tag{18}$$

$$\vdots$$

$$\theta_0 = \gamma - \theta_1 - \theta_2 - \cdots - \theta_K.$$

In conclusion, a general expression of $\theta_k$ in Eq (16) is expressed by

$$\theta_k = \begin{cases} \gamma(-\gamma)^k - \sum_{s=k+1}^{K} \theta_s \binom{s}{k} & \text{where } k \in \{0, 1, \cdots, K-1\} \\ w_0 (-\gamma)^k & \text{where } k = K \end{cases} \tag{19}$$

which is our desired objective.

## Experiments

We perform experiments to answer the following questions.

**Q1. Prediction Accuracy** How well does our MUSTAD preserve the multi-hop feature aggregation of a deep teacher model compared to other KD methods?

**Q2. Parameters vs. Performance** What is the trade-off between the number of parameters and the accuracy in student models?

**Q3. Ablation Study** How effectively do the multi-staged distillation and the single effective layer help a student conserve teacher's performance?

### Experimental setup

**Dataset.** We use four graph datasets as summarized in Table 2. Cora, Citeseer, and Pubmed are citation datasets where nodes and edges represent documents and citations, respectively. Each node feature indicates whether a word is included in each document. The ogbn-proteins dataset is an undirected and weighted graph where nodes represent proteins and edges mean different types of biological associations between proteins. An edge in the

**Table 2. Dataset statistics.**

| Dataset | Classes | Nodes | Edges | Features |
|---|---|---|---|---|
| Cora[1] [35] | 7 | 2,708 | 5,429 | 1,433 |
| Citeseer[1] [35] | 6 | 3,327 | 4,732 | 3,703 |
| Pubmed[1] [35] | 3 | 19,717 | 44,338 | 500 |
| ogbn-proteins[2] [36, 37] | 112 | 132,534 | 39,561,252 | 8 |

[1] https://linqs.soe.ucsc.edu/data

[2] https://ogb.stanford.edu/docs/nodeprop/

graph has an 8-dimensional feature, and a node has an 8-dimensional one-hot feature indicating which species the corresponding protein comes from.

**Teacher models.** We perform the distillation from two different teacher models. The first teacher model, GCNII [13] uses initial residual and identity mapping techniques and achieved state-of-the-art performance in Cora, Citeseer, and Pubmed. We compress the trained GCNII teacher in those three datasets. The second teacher model, GEN [8] proposes generalized message aggregators and pre-activation residual connections; GEN achieves a good performance in the ogbn-proteins dataset. We perform distillation from a trained GEN teacher in the ogbn-proteins dataset. When reproducing the teacher model, experimental settings such as data split, optimizer, regularization, activation functions, and hyperparameters follow those of [8, 13] unless explicitly stated.

**Competitors.** We compare MustaD with the following competitors:

- **KD** [17] is the model for knowledge distillation. It softens task predictions of the teacher and distills the knowledge of classes to a student. We denote the student distilled by this method as Student_KD. The final loss is:

$$\mathcal{L} = \mathcal{L}_{ce} + \lambda_{pred}\mathcal{L}_{pred} \tag{20}$$

- **LSP method** [21] distills an embedded topological structure of a teacher and achieved the best performance in graph-structured datasets. We denote the student trained by this method as Student_LSP. The final loss function is computed by:

$$\mathcal{L} = \mathcal{L}_{ce} + \lambda_{LSP}\mathcal{L}_{LSP} \tag{21}$$

All methods are implemented by PyTorch and PyTorch Geometric [38]. We use a machine with Intel E5-2630 v4 2.2GHz CPU and Geforce GTX 2080 Ti for the experiments.

## Semi-supervised node classification

**Cora, Citeseer, and Pubmed.** We perform distillation on trained GCNII models in Cora, Citeseer, and Pubmed. In particular, we perform KD from teachers with varying numbers of layers to show how well MustaD preserves the multi-hop feature aggregation of the teacher. When reproducing teacher models, we use the same settings as [13]. When training students, the early stopping patience is increased from 100 epochs to 200 epochs to get more stable results. Student_Base is a model trained with the ground truth labels without the teacher. Student_MustaD is our distilled student that has the same hidden feature dimension to the teacher. We train Student_KD with $\lambda_{pred}$ of 0.1, 0.1, and 100 on Cora, Citeseer, and Pubmed, respectively. For Student_LSP, we set $\lambda_{LSP}$ to 10 on both Cora and Citeseer. Student_LSP fails to be trained in Pubmed as every training node has only one neighbor which means there is no

**Table 3. Semi-supervised node classification accuracy for Cora, Citeseer, and Pubmed.** We perform the distillation from trained teachers with various number of GCN layers: 2, 4, 6, 8, 16, 32, and 64. Student_MustaD is our distilled student that has the same hidden feature dimension as the teacher. Note that MustaD consistently outperforms other KD methods while preserving the multi-hop feature aggregation of the deep teacher.

| Data | Model | Number of Parameters | Number of GCN Layers in the Teacher | | | | | |
|---|---|---|---|---|---|---|---|---|
| | | | **2** | **4** | **8** | **16** | **32** | **64** |
| Cora | Teacher [13] | **354K** (64 layers) | 81.83 | 82.92 | 84.13 | 84.56 | 85.28 | **85.34** |
| | Student_Base [13] | 96K | 79.71 | 79.71 | 79.71 | 79.71 | 79.71 | 79.71 |
| | Student_KD [17] | 96K | 80.05 | 80.12 | 80.31 | 80.54 | 80.76 | 79.41 |
| | Student_LSP [21] | 96K | 80.02 | 79.88 | 79.96 | 79.99 | 80.02 | 80.33 |
| | Student_MustaD | **96K** | 82.35 | 82.33 | 82.92 | 84.58 | 84.52 | **84.71** |
| Citeseer | Teacher [13] | **3,047K** (32 layers) | 67.62 | 68.13 | 70.77 | 72.87 | **72.89** | 72.71 |
| | Student_Base [13] | 1,015K | 67.82 | 67.82 | 67.82 | 67.82 | 67.82 | 67.82 |
| | Student_KD [17] | 1,015K | 68.21 | 68.03 | 68.35 | 68.92 | 68.87 | 69.06 |
| | Student_LSP [21] | 1,015K | 68.32 | 68.26 | 68.27 | 68.29 | 68.36 | 68.21 |
| | Student_MustaD | **1,015K** | 67.10 | 66.72 | 66.45 | 69.55 | 71.79 | **72.83** |
| Pubmed | Teacher [13] | **1,178K** (16 layers) | 78.59 | 77.94 | 78.13 | **80.35** | 79.95 | 79.96 |
| | Student_Base [13] | 195K | 75.61 | 75.61 | 75.61 | 75.61 | 75.61 | 75.61 |
| | Student_KD [17] | 195K | 75.71 | 75.87 | 76.01 | 76.03 | 75.84 | 75.98 |
| | Student_LSP [21] | 195K | - | - | - | - | - | - |
| | Student_MustaD | **195K** | 76.01 | 78.42 | 78.75 | 79.69 | 79.73 | **80.24** |

local structure to be distilled [13]. Other hyperparameters for each competitor are tuned to obtain the best results on the validation set. For our Student_MustaD, we set $\lambda_{pred}$ to 1, 0.1, and 100, $\lambda_{emb}$ to 0.01, 0.01, and 10 in Cora, Citeseer, and Pubmed, respectively, and the kernel function to KL divergence.

Table 3 shows the overall results on node classification in terms of mean accuracy after 50 runs. Note that our MustaD gives the best performance in terms of accuracy. In particular, Student_MustaD presents 3.77 ∼ 4.21%p improvement to the second-best model with 3.00 ∼ 6.04× smaller model size than the best teacher. Furthermore, the performance of the proposed MustaD increases as the number of layers in the teacher increases, unlike other KD methods. It indicates that MustaD preserves the aggregation process successfully whereas others do not. This also implies that MustaD gains more knowledge from the given input features when more GCN layers are used in the teacher.

In Citeseer and Pubmed, MustaD achieves the best performance when the student imitates 64 GCN layers of the teacher. However, the performance of the teacher decreases when more than 32 and 16 layers are stacked, respectively. It indicates that MustaD enables the student to aggregate information from farther nodes than the teacher does. If the accuracy of the teacher is too low, it is not easy for our student to show the remarkable performance consistently since MustaD aims to preserve the accuracy of deep teachers. However, the ability of MustaD to aggregate information from farther nodes than the teacher relieves the student's strong dependence on the performance of the teacher.

**ogbn-proteins.** We perform knowledge distillation using trained GEN teacher model in the ogbn-proteins dataset. Since the ogbn-proteins dataset is dense and large, full-batch training is not easy. We apply a random node sampler to generate batches for both mini-batch training and testing. Following [8], we set each batch size to one subgraph. Thus, as the number of batch increases, the size of the subgraph in each batch decreases and that leads to a

**Table 4. Multi-labeled node classification performance (AUC-ROC) in ogbn-protein.** The distillations are done from trained teachers with different numbers of GCN layers: 3, 7, 14, 28, and 56. Note that the proposed method Student_MUSTAD provides the best performance among the student models.

| ogbn-proteins | Number of Parameters | Number of GCN Layers in the Teacher | | | | |
|---|---|---|---|---|---|---|
| | | 3 | 7 | 14 | 28 | 56 |
| Teacher [8] | **483K** (28 layers) | 0.819 | 0.829 | 0.835 | **0.837** | 0.837 |
| Student_Base [8] | 42K | 0.797 | 0.797 | 0.797 | 0.797 | 0.797 |
| Student_KD [17] | 42K | 0.801 | 0.805 | 0.808 | 0.803 | 0.805 |
| Student_LSP [21] | 42K | 0.798 | 0.799 | 0.798 | 0.799 | 0.798 |
| Student_MUSTAD | **42K** | 0.811 | 0.819 | 0.821 | **0.823** | 0.820 |

decreased performance. Through experiments, we increase the number of batches from 10 to 40 to fit the large graph in our GPU (GeForce GTX 2080Ti with 11GB of memory) whereas [8] uses NVIDIA V100 with 32GB of memory. As a result, the reproduced teacher achieves the best performance with 28 layers although [8] achieves that with 112 layers. Without loss of generality, we perform the distillation on the reproduced teacher and validate our MUSTAD compared to other methods.

We evaluate the performance of each method on a multi-labeled node classification task in a semi-supervised setting. We train Student_KD with $\lambda_{pred}$ of 0.1 and Student_LSP with $\lambda_{LSP}$ of 10. For competitors, every hyperparameters are tuned to obtain the best results on the validation set. For our model, we set $\lambda_{pred}$ to 0.1, $\lambda_{emb}$ to 0.01, and the kernel function to KL divergence.

Table 4 summarizes the results in terms of AUC-ROC. MUSTAD presents an 1.55%p improvement in terms of AUC-ROC from the second-best KD model while requiring 11.41× fewer parameters than the teacher. Tables 3 and 4 show that our MUSTAD achieves the state-of-the-art performance with various teacher models.

## Parameters vs. performance

We perform a parameter study to show the trade-off between the number of parameters and accuracy. We vary the hidden feature dimension and the number of the effective layers in the student to vary the number of parameters. Furthermore, we vary the kernel functions used for distilling the knowledge of multi-hop feature representations and evaluate the performance. We analyze Cora with the trained 64-layered GCNII teacher.

**Hidden feature dimension.** We set the student's hidden feature dimensions to be the same as that of the teacher in the previous section; it limits the degree of model compression. We study the trade-off between the hidden feature dimension and the accuracy in Table 5. In particular, we vary the feature dimension from 16 to 128.

The table shows that Student_MUSTAD with the hidden feature dimension of 64 achieves the best performance. Note that setting the same feature dimension for the student as that of

**Table 5. Trade-off between the hidden feature dimension and the accuracy.** Note that the proposed MUSTAD with the hidden feature dimension of 64 shows the best performance.

| Hidden Feature Dimension | 16 | 32 | 64 | 128 |
|---|---|---|---|---|
| Number of Parameters | 24K | 49K | 96K | 193K |
| Accuracy (%) | 79.13 | 83.01 | **84.71** | 84.42 |

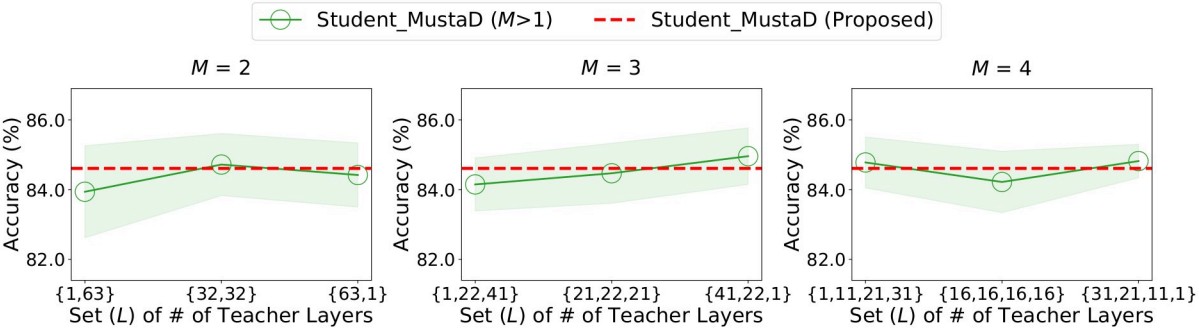

**Fig 3. Accuracy of MustaD for different numbers of the effective layers in a student.** $M$ represents the number of effective layers in the student, and $L$ corresponds to the set consisting of the number of teacher layers that each effective layer in the student imitates; e.g., $M = 2$ and $L = \{1, 63\}$ denotes that the two effective layer in the student imitates a GCN layer, and 63 GCN layers of the teacher, respectively. Note that MUSTAD, which has a single effective layer in the student, is enough to conserve the multi-hop feature aggregation of the teacher.

the teacher shows the best performance even when significantly smaller number of layers are used. It is also noteworthy that Student_MUSTAD with the hidden feature dimension of 32 still shows the best performance among KD methods shown in Table 3, while requiring 1.96× fewer parameters than the competitors, and 7.22× fewer parameters than the teacher. When the feature dimension is set to 128, MUSTAD shows a lower performance than MUSTAD with the dimension of 64, due to overfitting.

**Number of the effective layers.** Our proposed MUSTAD compresses the hidden GCN layers of a teacher into a single effective layer in a student. We study how the accuracy of MUSTAD changes as the number of the effective layer increases. However, to increase the number of the effective layer, we have to set the number of teacher layers that each effective layer in the student imitates. Let $M$ denote the number of the effective layers in the student. We tune the set $L$ consisting of the number of teacher layers that each effective layer in the student imitates in [{1, 63}, {32, 32}, {63, 1}] for $M = 2$, [{1, 22, 41}, {21, 22, 21}, {41, 22, 1}] for $M = 3$, and [{1, 11, 21, 31}, {16, 16, 16, 16}, {31, 21, 11, 1}] for $M = 4$.

Fig 3 shows that students having more than one effective layer shows a similar performance to the student having a single effective layer. It indicates that a single effective layer in the student is enough to conserve the multi-hop feature aggregation process of the teacher.

**Kernel functions.** MUSTAD uses various kernel functions (Eq 5) to distill the knowledge of multi-hop feature representations from the teacher. We compare students with different kernel functions in Cora and show the result in Table 6. We set $p$ to 2 for the distance based kernel. For the polynomial kernel, $c$ and $d$ are set to 2 and 0, respectively. For the RBF kernel, $\sigma$ is set to 1. In Table 6, 'None' represents the student model without the multi-staged knowledge distillation; i.e., it distills only the task prediction, not the embedding.

Note that the 'None' student shows a worse performance than students which distill embeddings with kernel functions. Among the kernel functions, KL divergence shows the best accuracy.

**Table 6. Accuracy with different kernel functions in the Cora dataset.** Note that KL divergence-based kernel provides the best accuracy, and the student 'None' without the embedding distillation shows a poor performance.

| Kernel Function | None | L2 Norm | Linear | Poly | RBF | KL Divergence |
|---|---|---|---|---|---|---|
| Accuracy (%) | 84.29 | 84.61 | 84.60 | 84.47 | 84.40 | **84.71** |

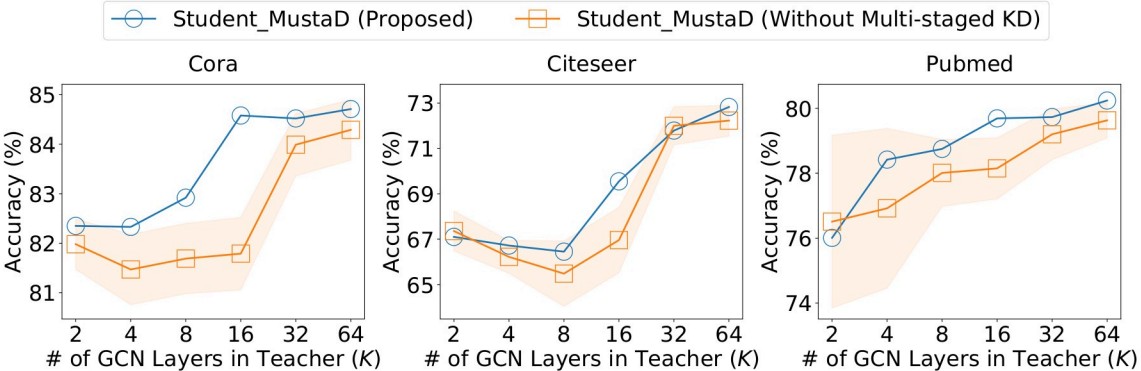

**Fig 4. Accuracy of MustaD without the multi-staged knowledge distillation.** Note that Student_MᴜsᴛᴀD (Without Multi-staged KD) is trained without distilling the knowledge of multi-hop feature representations; i.e., the teacher distills only the knowledge of task prediction to the student. MᴜsᴛᴀD with the distillation consistently shows a better performance compared to MᴜsᴛᴀD without it.

## Ablation study

We provide ablation studies for the effect of multi-staged knowledge distillation of the teacher, and the single effective layer in the student. The studies are done in three citation datasets; Cora, Citeseer, and Pubmed.

**Multi-staged knowledge distillation.** MᴜsᴛᴀD distills a teacher's knowledge in a multi-staged manner to conserve the accuracy. We show the effect of multi-staged knowledge distillation in Fig 4. Note that Student_MᴜsᴛᴀD (Without Multi-staged KD) is trained without distilling the knowledge of multi-hop feature representations; i.e., the teacher distills only the knowledge of task prediction to the student.

If we distill only the knowledge of task prediction, the teacher's error of prediction directly propagates to the student. However, the distillation of multi-hop features compensates for the error, and thus MᴜsᴛᴀD with the distillation presents a superior performance compared to MᴜsᴛᴀD without it as depicted in Fig 4. In other words, the multi-staged knowledge distillation takes a crucial role in acquiring proper knowledge from the teacher.

**Single effective layer.** MᴜsᴛᴀD imitates the multi-hop feature aggregation process of a teacher by a single effective layer. We investigate the effect of the single effective layer by comparing the proposed MᴜsᴛᴀD to a student with a single naive GCN layer.

Fig 5 shows that MᴜsᴛᴀD without the single effective layer presents significantly lower performance than the original MᴜsᴛᴀD. Furthermore, the accuracy of MᴜsᴛᴀD without the

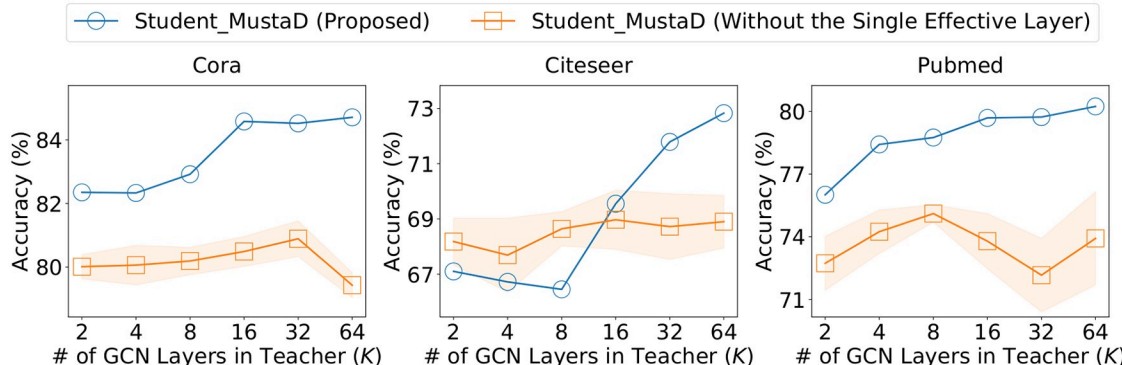

**Fig 5. Accuracy of MustaD without the single effective layer.** Note that the student without the single effective layer shows significantly lower performance than the original MᴜsᴛᴀD.

effective layer does not improve as the number $K$ of GCN layers in the teacher increases, whereas the performance of MustaD with that improves as $K$ increases. This is because the method preserves the teacher's multi-hop feature aggregation, which is the main purpose of the multiple layers in the teacher, by a single effective layer.

## Conclusion

In this work, we have proposed MustaD, an accurate method for compressing deep graph convolution networks (GCNs) by distilling multi-staged knowledge from a teacher. MustaD distills the teacher's knowledge of multi-hop feature aggregation by imitating the multiple GCN layers using a single effective layer in a student, which reduces the model size significantly, and by transferring the final hidden feature embeddings of the teacher to the student. MustaD also distills the knowledge of task prediction by transferring the prediction of the teacher. We give a theoretical analysis of MustaD, comparing the expressiveness of the proposed method to that of multi-layered GCN on a spectral domain. MustaD achieves the state-of-the-art performance in four real-world datasets, preserving the multi-hop feature aggregation of the teacher, compared to other distillation based GCN compression methods. Future works include extending MustaD to consider the semantics of features.

## Author Contributions

**Conceptualization:** Junghun Kim, Jinhong Jung.

**Data curation:** Junghun Kim, Jinhong Jung.

**Formal analysis:** Junghun Kim, Jinhong Jung.

**Funding acquisition:** U. Kang.

**Investigation:** Junghun Kim, Jinhong Jung.

**Methodology:** Junghun Kim.

**Project administration:** U. Kang.

**Resources:** Junghun Kim.

**Software:** Junghun Kim.

**Supervision:** U. Kang.

**Validation:** Jinhong Jung, U. Kang.

**Writing – original draft:** Junghun Kim.

**Writing – review & editing:** Jinhong Jung, U. Kang.

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
