## [Decision Letter · Decision Letter 0]

28 May 2021

PONE-D-21-15818

MustaD: Compressing Deep Graph Convolution Network with Multi-Staged Knowledge Distillation

PLOS ONE

Dear Dr. Kang,

Thank you for submitting your manuscript to PLOS ONE. After careful consideration, we feel that it has merit but does not fully meet PLOS ONE’s publication criteria as it currently stands. Therefore, we invite you to submit a revised version of the manuscript that addresses the points raised during the review process.

ACADEMIC EDITOR:

Based on the comments received from the reviewers and my own observation, I recommend major revisions for the paper. The authors should carefully address all the comments and suggestions from the reviewers. Also, the authors should proofread to polish the English grammar in the paper.

We look forward to receiving your revised manuscript.

Kind regards,

Thippa Reddy Gadekallu

Academic Editor

PLOS ONE

Journal Requirements:

2) Please amend either the abstract on the online submission form (via Edit Submission) or the abstract in the manuscript so that they are identical.

Reviewers' comments:

Reviewer's Responses to Questions

**Comments to the Author**

1. Is the manuscript technically sound, and do the data support the conclusions?

Reviewer #1: Yes

Reviewer #2: Yes

2. Has the statistical analysis been performed appropriately and rigorously? 

Reviewer #1: Yes

Reviewer #2: Yes

3. Have the authors made all data underlying the findings in their manuscript fully available?

Reviewer #1: Yes

Reviewer #2: Yes

4. Is the manuscript presented in an intelligible fashion and written in standard English?

Reviewer #1: Yes

Reviewer #2: Yes

5. Review Comments to the Author

Reviewer #1: 1. Does the proposed model has the ability to gain the full knowledge about all the features when there are multiple layers?

2. Does the author considers the semantic problems of features in the proposed method?

3. What happens to the Knowledge Distillation when the teacher’s accuracy is too low? Does the proposed system helps in solving these issues?

4. In the process of KD, Does the training errors in teacher propagate to student directly? Can the proposed system identify the errors?

Please cite the following papers

1. Ashokkumar P, Siva Shankar G, Gautam Srivastava, Praveen Kumar Reddy Maddikunta, and Thippa Reddy Gadekallu. 2021. A Two-stage Text Feature Selection Algorithm for Improving Text Classification. ACM Trans. Asian Low-Resour. Lang. Inf. Process. 20, 3, Article 49 (April 2021), 19 pages. DOI:https://doi.org/10.1145/3425781

2. G. Siva Shankar, P. Ashokkumar, R. Vinayakumar, Uttam Ghosh, Wathiq Mansoor, Waleed S. Alnumay, "An Embedded-Based Weighted Feature Selection Algorithm for Classifying Web Document", Wireless Communications and Mobile Computing, vol. 2020, Article ID 8879054, 10 pages, 2020. https://doi.org/10.1155/2020/8879054

Reviewer #2: - The quality of the figures can be improved more. Figures should be eye-catching. It will enhance the interest of the reader.

The abstract is long and NOT satisfactory. It should contain the following parts:

i. The importance of or motivation for the research.

ii. The issue/argument of the research.

iii. The methodology.

iv. The result/findings.

v. The implications of the result/findings.

- Please highlight the contribution clearly in the introduction

- In the first four paragraphs of literature review section, the authors have presented a good references, but they need to present the recent and most updated references.

- In the literature review section, you need to be consistent in the use of the verb tense, it is common to use the past tenses.

- The summary at the end of the literature review should be focused on the limitations of related work.

- The discussion is very important in research paper. Nevertheless, this section is short and should be presented completely.

- Major contribution was not clearly mentioned in the conclusion part.

- Make sure the Conclusion succinctly summarizes the paper. It should not repeat phrases from the Introduction!

- Authors should add the most recent reference:

1)  CANintelliIDS: Detecting In-Vehicle Intrusion Attacks on a Controller Area Network using CNN and Attention-based GRUCANintelliIDS: Detecting In-Vehicle Intrusion Attacks on a Controller Area Network using CNN and Attention-based GRU

2) DeepAMD: Detection and identification of Android malware using high-efficient Deep Artificial Neural Network, Future Generation Computer Systems 115, 844-856

6. PLOS authors have the option to publish the peer review history of their article (what does this mean?). If published, this will include your full peer review and any attached files.

Reviewer #1: No

Reviewer #2: No

---

## [Author Response · Author response to Decision Letter 0]

6 Jul 2021

Reviewer #1

(R1-1) Does the proposed model has the ability to gain the full knowledge about all the features when there are multiple layers?

– (A1-1) Accuracy of MustaD improves as the number of GCN layers in the teacher increases. This implies that MustaD gains more knowledge from the given input features when more GCN layers are used in the teacher. In other words, MustaD has the ability to gain the full knowledge about all the features when there are multiple layers in the teacher. We reported this in lines 334-336 in experiments section.

• (R1-2) Does the author considers the semantic problems of features in the proposed method? 

– (A1-2) Since the main purpose of MustaD is to compress a deep GCN model to a compact model while preserving the multi-hop feature aggregation of the deep model, it is hard to expect that MustaD considers the semantic problems of features. We added the discussion on this as a future work in conclusion section (lines 447-448).

• (R1-3) What happens to the Knowledge Distillation when the teachers accuracy is too low? Does the proposed system helps in solving these issues?

– (A1-3) Through the experiments summarized in Table 3, we have shown that MustaD has an ability to aggregate information from farther nodes than the teacher, thus relieving the student’s strong dependence on the performance of the teacher. We reported this in lines 341-345 in experiments section.

• (R1-4) In the process of KD, does the training errors in teacher propagate to student directly? Can the proposed system identify the errors?

– (A1-4) Fig 4 shows that MustaD compensates for the training error of merely propagating the prediction of the teacher. We discussed this in lines 420-425 in experiments section.

• (R1-5) Please cite the following papers: 1) Ashokkumar P, Siva Shankar G, Gautam Srivastava, Praveen Kumar Reddy Maddikunta, and Thippa Reddy Gadekallu. 2021. A Two-stage Text Feature Selection Algorithm for Improving Text Classification. ACM Trans. Asian Low-Resour. Lang. Inf. Process. 20, 3, Article 49 (April 2021), 19 pages. DOI:https://doi.org/10.1145/3425781, and 2) G. Siva Shankar, P. Ashokkumar, R. Vinayakumar, Uttam Ghosh, Wathiq Mansoor, Waleed S. Alnumay, ”An Embedded-Based Weighted Feature Selection Algorithm for Classify- ing Web Document”, Wireless Communications and Mobile Computing, vol. 2020, Article ID 8879054, 10 pages, 2020. https://doi.org/10.1155/2020/8879054

– (A1-5) We added the two references (lines 65-67 in related work section).

Reviewer #2

(R2-1) The quality of the figures can be improved more. Figures should be eye-catching. It will enhance the interest of the reader.

– (A2-1) We improved Figures 1 and 2. In particular, we redrew Fig 1 to be more informative, which previously looked complicated, and added markups for Fig 2 to be eye-catching.

• (R2-2) The abstract is long and NOT satisfactory. It should contain the following parts: 1) The importance of or motivation for the research, 2) The issue/argument of the research, 3) The methodology, 4) The result/findings, and 5) The implications of the result/findings.

– (A2-2) We rewrote the abstract to contain those five parts.

• (R2-3) Please highlight the contribution clearly in the introduction.

– (A2-3) We revised introduction section to emphasize the contribution of MustaD (lines 32-43 in introduction section). Furthermore, the contents of the contribution list have also been revised to clearly highlight the contributions (lines 54-56 in introduction section).

• (R2-4) In the first four paragraphs of literature review section, the authors have presented a good references, but they need to present the recent and most updated references.

– (A2-4) We added four up-to-date references (lines 65-67 in related work section).

• (R2-5) In the literature review section, you need to be consistent in the use of the verb tense, it is common to use the past tenses.

– (A2-5) We reviewed the literature review section to be consistent in the use of the verb tense.

• (R2-6) The summary at the end of the literature review should be focused on the limitations of related work.

– (A2-6) We added the limitations of deep GCNs at the end of the literature review (lines 104-106 in related work section).

• (R2-7) The discussion is very important in research paper. Nevertheless, this section is short and should be presented completely.

– (A2-7) We included more complete discussion (lines 334-336, lines 341-345, lines 380-387, lines 420-425, and lines 430-433 in experiments section).

• (R2-8) Major contribution was not clearly mentioned in the conclusion part.

– (A2-8) We revised the conclusion part to clearly present our major contribution.

• (R2-9) Make sure the Conclusion succinctly summarizes the paper. It should not repeat phrases from the Introduction!

– (A2-9) We revised conclusion section so that the conclusion succinctly summarizes the paper.

• (R2-10) Authors should add the most recent reference: 1) CANintelliIDS: Detecting In-Vehicle Intrusion Attacks on a Controller Area Network using CNN and Attention-based GRU, and 2) DeepAMD: Detection and identification of Android malware using high-efficient Deep Artificial Neural Network, Future Generation Computer Systems 115, 844-856.

– (A2-10) We added the two references (lines 65-67 in related work section).

---

## [Decision Letter · Decision Letter 1]

2 Aug 2021

MustaD: Compressing Deep Graph Convolution Network with Multi-Staged Knowledge Distillation

PONE-D-21-15818R1

Dear Dr. Kang,

We’re pleased to inform you that your manuscript has been judged scientifically suitable for publication and will be formally accepted for publication once it meets all outstanding technical requirements.

Kind regards,

Yuchen Qiu, Ph.D.

Academic Editor

PLOS ONE

Additional Editor Comments (optional):

Reviewers' comments:

Reviewer's Responses to Questions

**Comments to the Author**

1. If the authors have adequately addressed your comments raised in a previous round of review and you feel that this manuscript is now acceptable for publication, you may indicate that here to bypass the “Comments to the Author” section, enter your conflict of interest statement in the “Confidential to Editor” section, and submit your "Accept" recommendation.

2. Is the manuscript technically sound, and do the data support the conclusions?

Reviewer #2: Yes

3. Has the statistical analysis been performed appropriately and rigorously? 

Reviewer #2: Yes

4. Have the authors made all data underlying the findings in their manuscript fully available?

Reviewer #2: Yes

5. Is the manuscript presented in an intelligible fashion and written in standard English?

Reviewer #2: Yes

6. Review Comments to the Author

Reviewer #2: I would like to accept this paper now.

7. PLOS authors have the option to publish the peer review history of their article (what does this mean?). If published, this will include your full peer review and any attached files.

Reviewer #2: No

---

## [Editor Report · Acceptance letter]

5 Aug 2021

PONE-D-21-15818R1 

Compressing Deep Graph Convolution Network with Multi-Staged Knowledge Distillation 

Dear Dr. Kang:

I'm pleased to inform you that your manuscript has been deemed suitable for publication in PLOS ONE. Congratulations! Your manuscript is now with our production department. 

Kind regards, 

on behalf of

Dr. Yuchen Qiu 

Academic Editor

PLOS ONE